# X-ray 3D Imaging of Low-Density Laser-Target Materials

**Igor Artyukov** [1,*] **, Natalia Borisenko** [1] **, Gleb Burenkov** [2] **, Alexander Eriskin** [1] **, Maxim Polikarpov** [2] **and Alexander Vinogradov** [1,*]

1  P.N. Lebedev Physical Institite RAS, 53 Leninsky Prospekt, Moscow 119991, Russia; borisenkong@lebedev.ru (N.B.); eriskinaa@lebedev.ru (A.E.)
2  European Molecular Biology Laboratory, Hamburg Unit c/o DESY, Building 25A, Notkestrasse 85, 22607 Hamburg, Germany; gleb@embl-hamburg.de (G.B.); polikarpov.maxim@mail.ru (M.P.)
*  Correspondence: iart@lebedev.ru (I.A.); vinograd@sci.lebedev.ru (A.V.)

**Abstract:** Achieving optimal design and precise control of the internal structure of laser-target materials are the primary objectives in various laser physics experiments, particularly in generating high flux photon and neutron beams. The study of low-density materials poses considerable challenges for X-ray analysis due to their high transparency and minimal contrast. In this study, to obtain clear visualization of foams with sparse structures, we used phase-contrast X-ray tomography, utilizing a high-quality monochromatic X-ray beam from the synchrotron radiation source PETRA-III at DESY. Employing phase-contrast algorithms, the 3D structure of a foam-suspended glass microsphere inside the plastic cylinder was reconstructed with a level of image quality sufficient to visualize uniformity, displacement, and surface roughness on both sides of the microsphere. The primary focus of this investigation was a CH plastic capillary including 10 mg/cc CHO foam with a glass microsphere positioned at the center. The results of this study demonstrate that phase-contrast X-ray tomography with coherent synchrotron radiation is an effective and valuable technique for the development of new laser targets containing structured low-density materials.

**Keywords:** X-ray tomography; laser targets; synchrotron radiation; phase contrast; laser fusion; low-density materials; hohlraum

## 1. Introduction

A laser-fusion target typically consists of a solid microsphere for heavy isotopes of hydrogen and a cylinder hohlraum for converting laser light into X-rays and delivering energy to the hydrogen fuel [1,2]. An additional foam or aerogel layer plays an important role in compression stability control, energy transmission into the solid layers, and its transfer [3,4]. Creating these specific parts and materials is rather challenging, and there are no alternative sources to borrow them from [5,6]. The resulting fusion power drastically depends on the accuracy of the target design, fabrication, heating, and compression details. Therefore, the development and preparation of the targets require proper characterization procedures, which are still important and challenging tasks [7].

X-ray methods have traditionally been used to characterize the laser target as a whole and to measure a variety of its parameters. Among all of the methods of analysis, X-ray projection radiography is an old but still popular technique due to its simplicity, high speed of measurement, and ability to achieve the spatial resolution of several micrometers. To illuminate an object, X-ray radiation is typically emitted from a microfocus tube or other point-like sources, such as laser-driven plasma, Z-pinch, X-pinch, gas puff, or other discharges that can produce high-temperature and dense plasma [8,9].

Plasma focus (PF) sources are among the X-ray sources with the highest power and intensity of ionizing emission. They routinely generate neutrons with the energy of 2.5 MeV/14 MeV, X-rays ranging from 1–1000 keV, plasma jets, electron and ion

beams, and electromagnetic pulses. The duration of neutron and X-ray pulses is in tens of nanoseconds [10,11].

Another technique for the characterization of low-density (10 mg/cc) foam is based on the usage of multilayer X-ray mirrors. This involves transmission imaging with a laser plasma X-ray source produced by a rhenium target irradiated with an intensive laser beam. The plasma X-ray emission was selectively reflected and collimated at a wavelength of 4.5 nm by a Co/C multilayer spherical mirror and polyimide-backed Sc/C filter to produce images of relatively thick organic samples. This method has been proven capable of revealing the volume distribution of Cu nanoparticles (10 to 20 wt.%) within foam materials [12,13] but lacks spatial resolution and 3D reconstruction.

X-ray microtomography with laboratory X-ray sources is recognized as a highly effective method, which is suitable for the in situ examination of the final and intermediate laser-target products and related technological pathways [14,15]. The drawback of laboratory X-ray tomography is, however, a relatively long scanning time (of 10 to 30 hours) and the polychromatic spectrum of X-ray tube radiation. To cover for the aforementioned drawbacks, our work focuses on synchrotron-based X-ray microtomography [16–18], which is considered as one of the most promising methods for studying low-density materials and microstructures of laser targets, due to the ability to generate 3D sample scans with nm resolution in the time range of minutes. The main object of our investigation was to examine a microstructure intended for use as a target in laser fusion experiments. It comprises a small spherical fuel container embedded in low-density foam support within a polymer cylinder. This arrangement represents a physical model of the laser target inside a hohlraum.

Our previous routine laser-target X-ray diagnostics before the laser experiments have included microradiography using a soft X-ray microscope and laser-plasma emission [12,13] without 3D reconstruction. Additionally, to visualize the hydrogen fuel within the laser-target polystyrene microspheres, optical tomography was used at the P.N. Lebedev Physical Institute of RAS, but this technique is not suitable for studying opaque Be samples.

Here, we will present only the results of synchrotron X-ray 3D imaging, which is considered to be an essential extension of our previous diagnostics experiments.

## 2. Materials and Methods

### 2.1. Low-Density Materials in Laser Targets

The test objects under investigation in our study were different samples of laser targets containing low-density materials. These samples consisted of a spherical glass microcontainer for fusion fuel isotopes of hydrogen, which was securely placed inside a hollow plastic cylinder with a low-density three-dimensional network. It should be noted that the target components pose significant challenges for their analysis with conventional X-ray radiography. They exhibit significant variations in size, ranging from millimeters to nanometers, with a three-order magnitude difference in X-ray density.

There are several reasons to use low-density materials as laser targets. In recent years, a popular approach has been the use of near-critical aerogels for laser-initiated particle acceleration and electromagnetic field generation, also known as "table-top accelerators" [19–21]. In these applications, the essential processes are not sensitive to the surface, as electron acceleration occurs throughout the volume and requires sufficient time and path length inside the foam to achieve the prescribed energy. Therefore, a thorough investigation of foam uniformity throughout the volume using non-destructive methods is crucial before conducting laser experiments. We consider X-ray synchrotron computer tomography (CT) implemented with phase-contrast retrieval as one of the most efficient methods for the characterization of low-density foam materials and their corresponding laser targets.

The low-density material used in our study was trimethylolpropane triacrylate (TMPTA), produced as an aerogel with a bulk density of 10 mg/cc. This material is a 3D network containing many open pores with a size of approximately one micron. Throughout this work, we will refer to this aerogel as "foam", as commonly carried out in the literature.

The production method of such foam was developed at Dundee University (UK) and is described in [22].

In laser targets, the TMPTA foam can be employed to form an ablator on the surface of the spherical fuel shell, to study the plasma properties in planar layers, to mitigate plasma instabilities, to suspend the laser-target components, to provide laser particle acceleration, and so on. In our samples, the foam was used in a compound target to surround and fix a fuel microsphere inside a plastic cylinder.

The methods for the placement, alignment, and fixation of a gas-filled microsphere in polymer solution inside the cylinder are described in [23]. The microsphere's position is controlled with the help of micromanipulators and two orthogonal projections under a microscope. The plastic capillary cylinder is filled with the polymer solution containing the submerged microsphere. Once the desired alignment and position are achieved, the solution is irradiated with UV laser radiation, initiating polymerization of the solution and fixing the microsphere at the center. Subsequent processes involving non-solvent bath and supercritical drying processes result in a compound laser target consisting of the foam-filled cylinder with the microsphere embedded and centered within it.

In this study, we selected irregular samples of such compound laser targets to demonstrate the potential of synchrotron microtomography as a method for controlling and monitoring these microstructures. The foam and its structural elements (small fiber-like compactions, pores, and other inhomogeneity) were observed directly through the relatively dense plastic walls of the cylinder. As a result, our X-ray tomography experiments aimed to visualize not only the geometry and position of the target components but also the structure of the fine foam and the evenness of the surfaces.

Another sample (sample 2) in our study was CHO foam loaded with a uniformly distributed high-Z element (in this case, gold nanoparticles). These materials are of interest in laser fusion target production, as heavy chemical elements are efficient converters of laser radiation into X-rays and play an important role in driver energy transfer to the target. Additionally, they contribute to the stability of the laser plasma processes and the total fusion yield of the reaction [24].

To fabricate sample 2 for the synchrotron tomography experiments, we followed the procedure described in [25]: the CHO polymer containing 0.05 wt.% of gold atoms in the form of water-solvable salt underwent UV photo-initiated synthesis, non-solvent bath treatment, and supercritical drying. It should be mentioned that the common pink color of the produced CHO foam is a characteristic feature of the gold colloid solution.

*2.2. Phase Contrast X-ray Tomography with the Synchrotron Radiation*

To examine the laser-target internal structure, we utilized a synchrotron radiation beamline that provided a high-quality monochromatic X-ray beam needed for phase-contrast imaging. The experiments were carried out on the P14 beamline at synchrotron PETRA III (DESY, Hamburg, Germany) [26]. This X-ray beamline offered a photon flux exceeding $10^{13}$ photon/s within a field of view of $0.6 \times 1.2$ mm$^2$. The transverse coherence of the beam was about 500 μm and 20 μm at the sample position in vertical and horizontal directions, respectively, providing opportunities for phase-contrast X-ray imaging. This beamline setup has been successfully used for various advanced X-ray diffraction experiments, e.g., for the fast high-resolution phase-contrast X-ray tomography of large macromolecular crystals (>100 μm) [27].

The samples were mounted on a needle holder and illuminated by the X-ray beam. To collect the tomographic data, the holder was fixed on a stage that enabled precise rotation with an error margin of less than 0.001 degrees.

X-ray images were acquired using an optical system consisting of an 8 μm-thick LSO:Tb scintillator, 20× microscope, and 2048 × 2048 pco.edge 4.2 sCMOS camera (Excelitas PCO GmbH, Kelheim, Germany). Considering the microscope magnification, the effective pixel size on the object was estimated to be 0.325 μm over the field of view of about 600 μm.

A typical tomographic scan included capturing 30 flat-field images and 3600 projections with a step of 0.1°. Flat-field correction was applied by dividing each projection image by the most similar flat-field image selected based on the structural similarity index measure (SSIM) criterion [28]. To account for possible axis shift at different object-camera distances, the X-ray images were corrected using the Fourier-space correlation with a sub-pixel interpolation [26].

To investigate objects with low X-ray absorption contrast we used the technique of propagation-based phase-contrast imaging [29,30]. The technique treats a change in the wave intensity patterns during free propagation as a result of the Fourier transform, whereby the spatial frequencies are defined by the distance of wave propagation. By combining the equations for amplitude and phase and by solving them through a least-squares optimization procedure, it is possible to describe the phase shift inside the sample and visualize boundaries and interfaces between materials with different refractive indices. To achieve higher image quality, in our study, the tomographic scan of the same sample was repeated four times at different distances (135–145 cm) between the sample and the X-ray detector. Four sets of recorded X-ray projection images were then processed using a multi-distance non-iterative holographic reconstruction method, resulting in the restoration of one high-contrast projection image per each step of rotation. After that, tomographic reconstruction was performed using the TOMOPY package [31] with the application of the Gridrec algorithm and Shepp–Logan filtering.

To enhance the presentation of the reconstructed data in the final 3D views, the following image processing software was used: Fiji-ImageJ 1.53 t (open code) and CTvox v. 3.3.0 r1403 (Bruker MicroCT, Kontich, Belgium). The Fiji-ImageJ program allowed us to improve the brightness/contrast parameters of the reconstructed images. The whole stack of the sections was visualized as a 3D object using the CTvox program, which enabled highlighting and adjustment of the local intensity and different densities. The results of X-ray phase-contrast CT imaging, 3D reconstruction, and visualization are given and discussed in Section 3.

## 3. Results

### 3.1. Microsphere in a Polymer Cylinder

The first analyzed sample was a low-density structure within the laser target: a foam-filled cylinder with a microsphere centered inside. The X-ray projections were recorded at the photon energy of 18 keV with an exposure time of 10 ms per frame.

Figure 1 shows the reconstructed rendered 3D view (CT image) of Sample 1. One can observe the symmetrical positioning of the outer cylinder and the inner fuel container (microsphere). The solid walls of the microsphere indicate no damage. Figure 1 also highlights the uniformity of the surrounding foam within this delicate structure throughout the whole volume, with the foam density being significantly lower than that of the glass sphere.

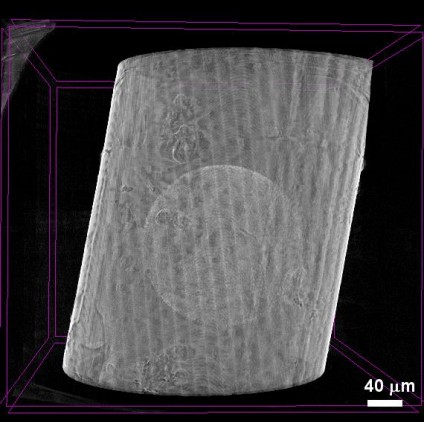

**Figure 1.** 3D general view of the fusion fuel glass container suspended with the nanostructured low-density foam inside the plastic cylinder.

As was mentioned above, in our study, we investigated only non-ideal samples of the laser targets, which were not used in laser-plasma experiments. On the other hand, visualization of the micro-structure imperfections is considered to be an evident demonstration of the efficiency of X-ray synchrotron microtomography as a method for such kinds of investigations.

Figure 2 shows that the fuel container successfully endured the technological procedure of laser-target fabrication. This microsphere withstood wet processing, numerous handling and mounting steps, and critical point drying. It is important to note that, for better visibility of the foam in Figure 2, the transparency of the cylinder walls was enhanced in the image.

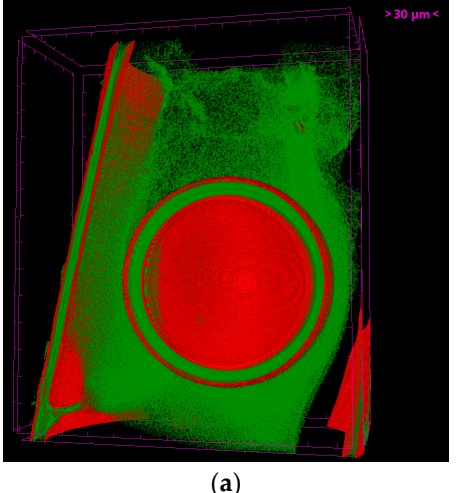
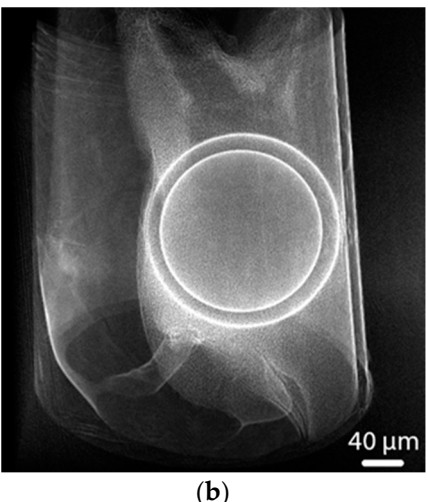

(a)                                                                                     (b)

**Figure 2.** (**a**) 3D virtual plane cut is used to monitor the sphericity of the inner fusion fuel container and the thickness of the cylinder walls; (**b**) The fusion fuel glass container and low-density foam presented with an attenuated view of the cylinder.

In Figure 2a, a less dense substance can be observed at the top of the picture, while a denser material is situated below the microsphere and to the right of it. One can also observe no damage and the perfect sphericity of the fuel microsphere, with the uniform thickness of its wall being 15 µm. The thickness of the cylinder wall was measured to be 11 µm. Figure 2b clearly indicates the decentralized location of the spherical fuel container. In addition, there are cracks and density uniformities observed in the foam-filled parts of the volume.

As was mentioned previously, our study originally examined imperfect samples of laser targets that were not used in laser-plasma experiments. However, visualizing the imperfections in the microstructure serves as a clear demonstration of the effectiveness of X-ray synchrotron microtomography.

It is also important to highlight that X-ray synchrotron tomography has demonstrated a dynamic range high enough for the simultaneous observation of the low-density CHO foam with a density of 10 mg/cc, the inner glass spherical fuel container with a density of 2500 mg/cc, and the plastic CH cylinder with a density of about 1000 mg/cc.

Figure 3 serves as confirmation of the microsphere's suitability as a possible fuel container. In the lower left part of Figure 3a, a distinct surface defect resembling a crack is visible on the microsphere's surface. Naturally, a through-hole crack would be a severe defect, resulting in gas leakage from the fuel container, which makes the container unsuitable. However, as seen in Figure 3b, no cracks or holes are observed in the internal surface. Thus, we can infer that the object shown in Figure 3a appears to be a stick-like piece of another material, such as a broken glass shell, which ended up stuck to the external surface of the fuel microsphere without any serious damage to the surface.

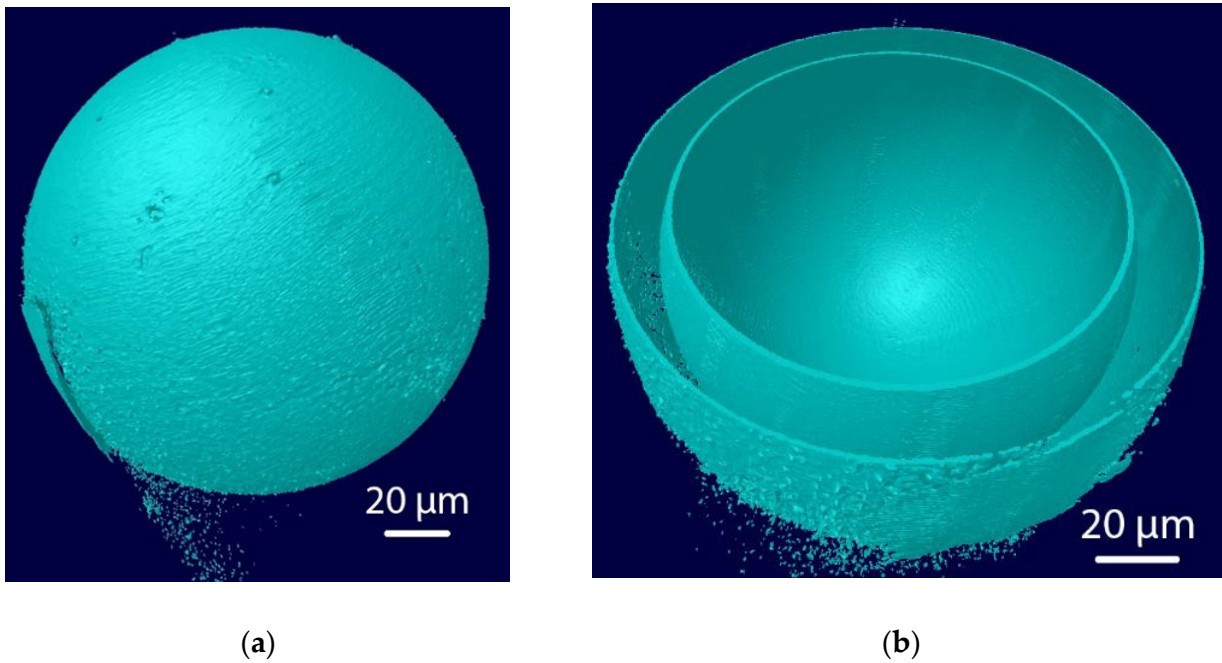

(**a**) (**b**)

**Figure 3.** (**a**) 3D external view of the glass microsphere shown without surrounding materials; (**b**) Presentation of the internal surface of the glass microsphere.

Figure 4a shows the sectional views of sample 1. To ensure the visibility of the cross-sections, the plane above the microsphere and foam is slightly dimmed. The foam is depicted as a partially transparent medium in order to represent the inner microsphere. Figure 4b exhibits the cross-section of the foam, cut along the microsphere. As was mentioned above, the optically thinner material is located at the top part, while the optically denser material accumulates at the bottom and to the right of the microsphere. The cross-section of the fuel microsphere shows its good sphericity in all directions, with a uniform shell wall thickness of 15 μm.

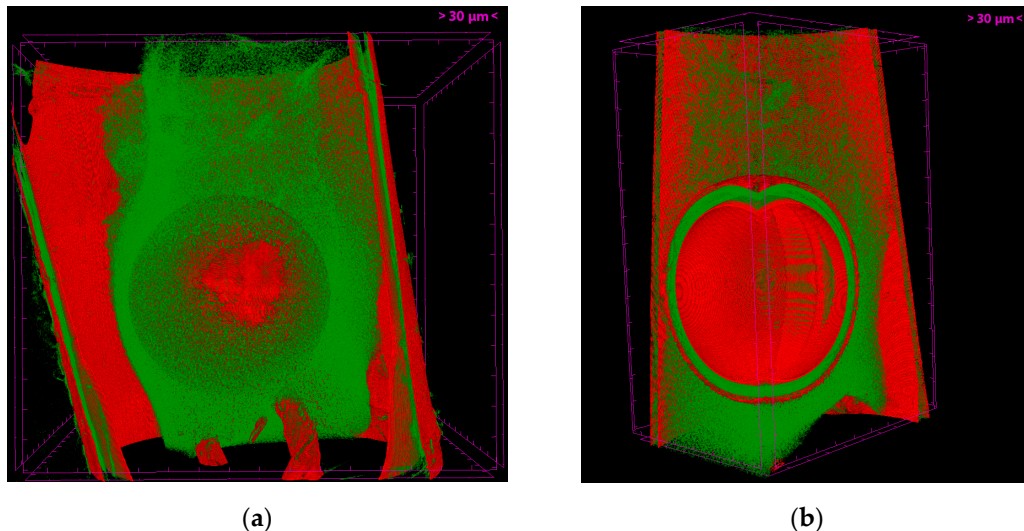

(**a**) (**b**)

**Figure 4.** (**a**) Side view of the object as seen previously from Figure 2a; (**b**) The foam-suspended sphere in the cylinder is cut virtually by two perpendicular planes.

### 3.2. CHO Foam Loaded with High-Z

The second type of laser-target material investigated was a CHO foam loaded with high-Z elements. In this case, a 0.05% gold admixture was uniformly added to the foam all across the volume of sample 2. The tomographic scans were recorded on the P14 beamline

at the photon energy of 18 keV. Given the extremely low density of the foam material in this case, it was crucial to enhance the contrast in the X-ray images with the help of phase-contrast X-ray imaging. The scans were performed at four different distances between the camera and the sample (4.5 cm, 5 cm, 5.6 cm, and 6.5 cm) with optimized values to obtain high (but not yet saturated) phase contrast, reconstructed with the help of phase retrieval algorithms as described in Section 2.

Achieving uniformity in the spatial distribution and sizes of metal nanoparticles when adding and mixing them into the foam volume is crucial. Chemical reduction of metallic salts with the formation of colloidal metal nanoparticles is a frequently used method of solid or gel M-loading (Au in the CHO polymer).

In sample 2 (Figure 5), the 0.05% gold admixture is assumed to be uniformly dispersed throughout the sample volume. However, the obtained CT image (see Figure 5) also reveals the defects resulting from the variation in the density caused by possible fluctuation in the local concentration of the colloidal gold particles. This effect is typically attributed to uncontrolled microfluidic phenomena during the fabrication of the sample.

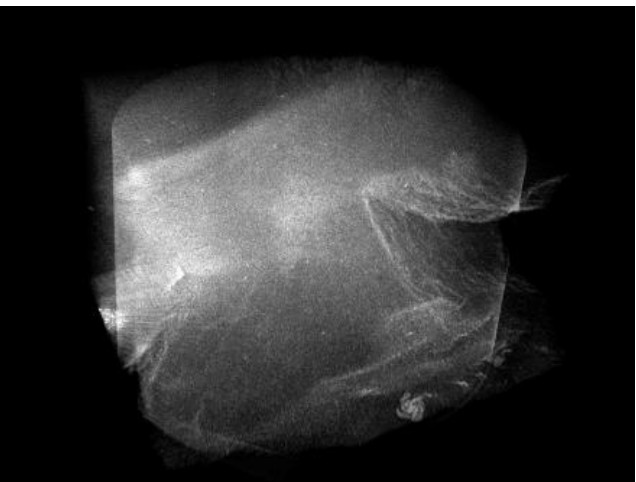

**Figure 5.** 3D view of the CHO foam loaded with high-Z.

Figure 5 provides clear visualization of density variation and uniformity in low-density foams, which have garnered significant interest in recent research on laser particle acceleration in low-density media [32,33].

It is important to note that the application of these laser targets extends beyond plasma production experiments. With higher metal concentrations, gold serves as an excellent converter of laser light into X-rays. The targets and their characterization are equally relevant in laboratory astrophysics experiments, the study of non-linear phenomena, the mitigation of plasma instabilities, laser–plasma interaction, and others.

## 4. Discussion

In this study, we have demonstrated the applicability of synchrotron-based phase-contrast imaging to the structural analysis of non-fusion laser targets. The target configuration of the first sample consisted of a glass microsphere container for hydrogen fuel and a low-density support foam inside a hollow plastic cylinder. The glass micro balloon of laser-target quality was fabricated beforehand at the Lebedev Physical Institute using the original method. [34].

The primary objectives of this study were as follows:

- To determine the feasibility of characterizing the low-density material 3D aerogel network), which is only a few times denser than air under normal conditions;
- To assess the visibility and reconstruction capabilities of the nanostructured aerogel network within the solid parts of the target assembly;

- To demonstrate the potential of synchrotron tomography for characterizing the entire target, encompassing all its constituents simultaneously.

The tomographic reconstruction of the samples allowed us to reveal and identify various characteristic details. As the densities (both mass and optical) of these parts differed by at least three orders of magnitude, synchrotron X-ray tomography allowed us to successfully reconstruct all components of the studied object, enabling precise measurements of the complex microstructured assemblies.

Sample 1 involved three parts: the microsphere, the foam, and the cylinder. These parts were not mechanically assembled, and they were not fabricated independently. Instead, the microsphere and cylinder underwent separate technological processes through several stages of synthesis, treatment, aging, and critical point drying techniques. Phase-contrast X-ray microtomography helped us to visualize the internal structure and quality of all three parts, due to their significant differences in material density. Thus, the presented approach enabled non-destructive characterization of the target assembly, including its geometry, prescribed composition, construction symmetry, and design suitability.

We believe that the detailed analysis of laser targets cannot be accomplished without phase-contrast measurements when the inner layer of condensed fuel is present in the target for ICF (inertial confinement fusion) or IFE (inertial fusion energy) experiments. Elements such as beryllium and other light elements required for energy-efficient fusion burn can only be accurately studied using the presented X-ray phase-contrast approach. Such targets exhibit a hierarchy of scales and densities similar to the relative parameters explored in the present study [35].

**Author Contributions:** Conceptualization, I.A. and N.B.; methodology, I.A., N.B, M.P. and G.B.; software, M.P. and G.B.; validation, N.B. and A.E.; formal analysis, I.A.; investigation, M.P., G.B., N.B. and I.A.; resources, G.B. and M.P.; data curation, M.P.; writing—original draft preparation, I.A., M.P. and N.B.; writing—review and editing, I.A. and A.V.; visualization, I.A. and A.E.; supervision, A.V.; project administration, I.A.; funding acquisition, I.A., N.B. and A.V. All authors have read and agreed to the published version of the manuscript.

**Funding:** This research was partly funded by the P.N. Lebedev Physical Institute New Scientific Group 55.

**Acknowledgments:** The support with the experimental chemistry given by W. Nazarov and the support with the sample preparation and treatment given by L. Borisenko are gratefully acknowledged.

**Conflicts of Interest:** The authors declare no conflict of interest.

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
