# Peer review of "X-ray 3D Imaging of Low-Density Laser-Target Materials"

_photonics, doi:10.3390/photonics10080875_

Round 1
Reviewer 1 Report
Dear Authors,
This paper reports the findings of a study for the development of new laser targets for the generation of high-flux photon and neutron beams. This work presents a method for imaging the internal structures of low-density materials using high-quality X-ray beams and also applying phase contrast analysis algorithms. The results show that high-quality X-ray radiation and tomographic imaging based on phase contrast analysis is an effective and useful technique for the development of laser targets containing newly structured low-density materials. The study provides an effective method for imaging the internal structures of low-density materials using high-quality X-ray beams and phase contrast analysis. This provides a new approach to the development and characterization of laser targets. The work focuses on imaging the internal structures of low-density materials with high transparency and barely perceptible contrast. The suitability of such materials for X-ray analysis and a detailed examination of their internal structures contribute to the literature.
Making the following corrections will improve the quality of the work.
1. Although the study provides an effective method for visualizing the internal structures of low-density materials, a more comprehensive characterization can be done.
2. Studies on different material types by increasing the material samples for work can present a broader perspective to the literature.
3. The findings of the study show that tomographic imaging based on high-quality X-ray beams and phase contrast analysis is an effective technique for the development of laser targets. This technique is important for high-flux photon and neutron beam generation. However, it would be more appropriate to give more academic technical information to the material part of the article for the method and analysis processes in the study. Results obtained using different phase contrast algorithms or data processing techniques should be compared. In addition, more generally valid results can be obtained by increasing the variety of samples and considering different experimental conditions.
4. It is necessary to compare the results of this study with other similar methods or alternative techniques.
5. There are grammatical and spelling errors, and the entire article should be checked.
Regards.
There are grammatical and spelling errors, and the entire article should be checked.
Reviewer 2 Report
Dear Authors,
Please read the attached pdf document to find my Review of Your Paper.

The paper is well written.
Round 2
Reviewer 1 Report
Necessary corrections have been made and are acceptable.
Necessary corrections have been made and are acceptable.
Author Response
Dear Reviewer,
We made an intensive correction of our English in the text. The revised manuscript is submitted with all changes tracked by WORD reviewer track tool.
Thank you again for your positive estimation of our work.
Sincerely,
Igor Artyukov
